# A Combined Cell-Worm Approach to Search for Compounds Counteracting the Toxicity of Tau Oligomers In Vivo

**DOI:** 10.3390/ijms231911277

**Published:** 2022-09-24

**Authors:** Carmina Natale, Maria Monica Barzago, Luca Colnaghi, Ada De Luigi, Franca Orsini, Luana Fioriti, Luisa Diomede

**Affiliations:** 1Department of Molecular Biochemistry and Pharmacology, Istituto Di Ricerche Farmacologiche Mario Negri IRCCS, Via Mario Negri 2, 20156 Milan, Italy; 2Dulbecco Telethon Institute and Department of Neuroscience, Istituto Di Ricerche Farmacologiche Mario Negri IRCCS, Via Mario Negri 2, 20156 Milan, Italy

**Keywords:** tauopathy, tau protein, oligomers, proteotoxicity, *C. elegans*, tetracyclines

## Abstract

A clear relationship between the tau assemblies and toxicity has still to be established. To correlate the tau conformation with its proteotoxic effect in vivo, we developed an innovative cell-worm-based approach. HEK293 cells expressing tau P301L under a tetracycline-inducible system (HEK T-Rex) were employed to produce different tau assemblies whose proteotoxic potential was evaluated using *C. elegans*. Lysates from cells induced for five days significantly reduced the worm’s locomotor activity. This toxic effect was not related to the total amount of tau produced by cells or to its phosphorylation state but was related to the formation of multimeric tau assemblies, particularly tetrameric ones. We investigated the applicability of this approach for testing compounds acting against oligomeric tau toxicity, using doxycycline (Doxy) as a prototype drug. Doxy affected tau solubility and promoted the disassembly of already formed toxic aggregates in lysates of cells induced for five days. These effects translated into a dose-dependent protective action in *C. elegans*. These findings confirm the validity of the combined HEK T-Rex cells and the *C. elegans*-based approach as a platform for pharmacological screening.

## 1. Introduction

After mutation or post-translational modification, particularly phosphorylation, the microtubule-binding protein tau dissociates from microtubules and undergoes aggregation, a process that begins with the formation of soluble aggregates and ends with the deposition in the cytoplasm of cells of large fibrillar hyperphosphorylated aggregates called neurofibrillary tangles (NFTs) [1]. This aggregation process is one of the well-established mechanisms underlying a class of neurodegenerative diseases called tauopathies [1]. Although NFTs are a key pathological hallmark of these diseases, in animal models of tauopathy, their presence did not correlate with the neuronal loss and memory decline associated with the disease [2]. Extracellular tau, in the form of oligomers more than fibrillar assemblies, has instead been proposed as the main cause of the onset and progression of tauopathies [3].

Based on evidence from different cell-based and animal studies, tau oligomers were proposed as the main cause of the toxicity. Oligomers obtained from sonicated fibrils, but not fibrils themselves, reduced the viability of human neuroblastoma SH-SY5Y cells and their toxicity was counteracted by incubation with tau oligomer-specific T22 antibody [4,5]. In neurons derived from induced pluripotent stem cells, tau oligomeric assemblies caused neurite degeneration, neuronal loss, and alterations of synaptic transmission and neuronal activity [6]. Tau oligomers can also be internalized by hippocampal neurons, inducing functional alterations and reducing cell viability [7].

Tau oligomers with apparent molecular weights of 140 kDa and 170 kDa were found in the brain of two mouse models of tauopathy, both expressing human tau carrying P301L substitution [8]. This mutation, associated with frontotemporal dementia (FTD), results in a protein more prone to aggregate and became hyperphosphorylated than the wild type [9,10,11,12]. In these animals, the cerebral level of oligomers correlated well with the memory impairment associated with the disease progression [8].

Similar tau oligomeric species were found in the cerebral tissues of patients with Alzheimer’s disease (AD) and FTD with parkinsonism linked to chromosome 17, underlying their clinical significance [8]. In addition, the injection of tau oligomers, but not monomers or fibrils, into the brain of C57BL/6 mice caused a memory deficit comparable to that observed in mouse models of AD and was accompanied by neuronal death, synaptic dysfunctions, and alterations in mitochondrial function [13].

Among the extracellular multimeric forms of tau, trimers were suggested as the minimum assembly required for cellular uptake, seeding, and toxicity in different cell lines and primary neurons [14,15]. However, a clear relationship between the precise tau assembly/assemblies and toxicity remains to be established. Their identification is no trivial matter since they are a priority for the development of new therapeutic strategies, still needed today.

To correlate the proteotoxic effect with the molecular tau assemblies in vivo we developed an innovative approach involving HEK293 cells expressing tau P301L under a tetracycline-inducible system (HEK T-Rex) and the nematode *Caenorhabditis elegans*. HEK T-Rex cells have already been used in vitro to correlate intracellular tau aggregation and cell viability [16,17]. More recently, HEK T-Rex cells expressing tau P301L tagged with a green fluorescent protein (GFP) have been used as a cell-based tau overexpression assay to screen for pharmacological compounds against tau aggregation [18].

Here, we assessed the proteotoxic potential of the various tau assemblies produced by HEK T-Rex cells at different times after the induction, using *C. elegans* as a biosensor, which we have previously shown can specifically recognize the toxicity of amyloidogenic proteins, including tau [19]. We recently demonstrated that the administration of (1) recombinant tau oligomers, (2) soluble tau assemblies produced in the brain of P301L transgenic mice, and (3) tau species formed in the brain of mice after traumatic brain injury (TBI) [20,21] impaired the worm’s motility and synaptic transmission.

We also examined whether this cell-worm-based approach can be employed as an experimental platform to search for pharmacological agents to counteract the toxicity of tau oligomers. To this end, we tested the effect of doxycycline (Doxy), a tetracycline with a known antibiotic activity that also has pleiotropic effects against various amyloidogenic proteins [22,23,24,25]. Doxy was recently reported to reduce amyloid aggregation of recombinant tau, prevent tau seeding and lower the toxicity of tau aggregates in vitro [26]. Here, we demonstrate for the first time that Doxy protected against the proteotoxic effect of tau oligomers in vivo.

This reinforces the rationale for repurposing this drug as an anti-tauopathy agent and confirms the validity of the combined HEK T-Rex cells and *C. elegans*-based approach as a platform for pharmacological screening.

## 2. Results

### 2.1. Multimeric Tau Assemblies from HEK T-Rex Are Proteotoxic for C. elegans

We recently demonstrated that *C. elegans* recognized the tau as toxic in brain homogenates from vertebrate models of tauopathy [21]. We, therefore, applied this approach to investigate the characterization of the tau assemblies responsible for the proteotoxic effect in vivo. As the source of tau, we employed HEK T-Rex cells expressing, under the control of a tetracycline-inducible system, the human P301L mutated protein fused to a HA tag. Thus, cells were treated with 1 µg/mL Doxy to induce the expression of tau P301L (Induced) (Figure 1a) and harvested immediately before (time 0) and 1, 3, and 5 days later. Control cells were treated with the same volume of 10 mM phosphate-buffered saline (PBS), pH 7.4, and harvested at the same time points (Not-induced) (Figure 1a). The induction of tau expression did not affect the viability of Induced cells at any of the time points (Appendix A).

Cell lysates from Not-induced and Induced cells were then administered to *C. elegans* and the locomotor activity of nematodes was evaluated seven days later by counting the number of body bends/min (Figure 1b). Control worms were treated with the same volume of 10 mM PBS, pH 7.4 (Vehicle). Motility was not affected in nematodes treated with lysates of Not-induced cells and cells induced for 0, 1, and 3 days compared to controls (Figure 1b). Only the treatment of nematodes with cell lysates from HEK T-Rex cells induced for 5 days significantly reduced the locomotor activity compared to Not-induced cells at the same time points and Vehicle-fed worms (Figure 1b).

Experiments were then performed to correlate the onset of the locomotor impairment in worms with the amount and conformational state of tau P301L expressed by cells. Western blot analysis was carried out first to record the level of total tau and phosphorylated tau (P-tau) in lysates of Not-Induced and Induced cells harvested 1, 3, and 5 days after induction. As shown in Figure 2, similar amounts of tau and P-tau were produced in Induced cells from 1 to 5 days of induction whereas no tau and P-tau were detected at time 0 (Appendix A), indicating that neither the level of the protein nor its degree of phosphorylation correlated with the toxicity in the worms.

To confirm the specific role of tau in the toxic effect in *C. elegans*, lysates of Not-Induced cells and cells Induced for 5 days were immunodepleted of tau before administration to worms (Figure 3). The tau P301L expressed by HEK T-Rex cells were tagged with HA, so tau in the lysates was immunoprecipitated by incubation with an anti-HA tag antibody [27]. Cell lysates were incubated in the same experimental conditions with an anti-GFP antibody as negative control [28]. Lysates were analyzed by Western blot before (Appendix A) and after the immunoprecipitation to confirm tau depletion (Figure 3a,b) and then given to *C. elegans*. Tau immunodepletion abolished the motility defect caused by the lysates from cells induced for 5 days (Figure 3c), indicating that tau had a key role in the toxic effect in *C. elegans*.

We then examined whether this toxic effect could be related to changes in the solubility of the tau P301L expressed. Detergent solubility assays were performed on lysates of cells induced for 3 and 5 days to determine the levels of soluble and insoluble tau assemblies. After 5 days of induction, the cells contained a significantly higher percentage of insoluble tau than cells induced for 3 days (Figure 4), indicating that the toxicity in worms may be related to a change in protein solubility.

To gain information on the characterization of tau assemblies responsible for toxicity, lysates of cells induced for 3 and 5 days were subjected to Western blot analysis using a T22 antibody, to specifically recognize oligomeric tau [4]. Oligomers were detected in lysates of cells induced for 3 days and their levels were significantly higher in lysates of cells induced for 5 days (Figure 5a,b). This observation was further supported by semi-denaturing 8% SDS-PAGE gel analysis followed by Western blot (Figure 5c,d). Lysate of cells induced for 3 days had a more intense tau monomer band at 63 kDa than the cells induced for 5 days, which showed an increase of the immunoreactive signal corresponding to the oligomeric molecular weight tau species (Figure 5c). Separation on a native PAGE 3–8% gradient gel indicated the tau oligomeric bands at ~180 kDa in the lysates of cells induced for 3 days and at ~240 kDa in the lysates of cells induced for 5 days (Figure 5d).

These findings indicate that the expression of tau P301L by HEK T-Rex cells resulted in the formation of multimeric tau assemblies with time and suggested that only those with molecular weight resembling the tetrameric protein mediate the proteotoxic effect in *C. elegans*.

### 2.2. Doxy Protected from the Toxicity Induced by Multimeric Tau

We investigated whether *C. elegans* can be employed to test compounds protecting against oligomeric tau toxicity. We tested the effect of Doxy which, at 100 µM, prevented the seeding of tau and counteracted the toxicity of aggregates in SH-SY5Y cells [26]. Although we used Doxy in our experimental condition to induce tau expression in HEK T-Rex cells, the concentration of 1.95 µM we used was certainly lower than that used for pharmacological studies. Furthermore, for the *C. elegans* experiments, we used lysates from cells collected five days after the induction in which the residual Doxy concentration is likely to be close to zero. Lysates of HEK T-Rex cells in which the tau expression was induced for five days were incubated for 2 h with different concentrations of Doxy before being administered to the worms.

Doxy protected worms in a dose-dependent manner from the toxic effect of cell lysate, with a half-maximal inhibitory concentration (IC50) of 8.06 µM (Figure 6a). At the optimal concentration of 50 µM, Doxy completely abolished the neuromuscular impairment induced by cell lysates in the worms (Figure 6b).

The protective effect of Doxy cannot be ascribed to an effect on the total levels of tau and P-tau in cell lysates (Appendix A) but to its ability to reduce the level of insoluble tau, as indicated by the detergent insolubility assay (Figure 6c,d). Analysis of samples under semi-denaturing conditions indicated that Doxy reduced the intensity of the immunoreactive signals corresponding to the tau monomeric band at 63 kDa as well as tau oligomeric bands at ~240 kDa, responsible for the toxicity (Figure 6e). From these data, we concluded that Doxy exerts a protective effect against the toxicity induced in *C. elegans* by tau oligomers, affecting their conformational state.

## 3. Discussion

We developed a new experimental approach based on the combined use of HEK T-Rex cells and *C. elegans* to produce multimeric tau assemblies and demonstrate their toxicity in vivo. HEK293 cells expressing full-length tau isoforms under the control of tetracycline have already been employed to investigate the mechanisms of protein aggregation and the involvement of phosphorylation in conformational changes [17,18]. We show that the overexpression of tau in these cells for 1 up to 5 days did not result in any significant change in the phosphorylation status of the protein but caused a time-dependent formation of multimeric insoluble assemblies. These findings indicate that HEK T-Rex, already employed for studies on tau fibrillization, can also be used to study the first phases of aggregation. The use of cells instead of recombinant protein as the source of tau offers the advantage of avoiding extraction and purification methods for the isolation of tau oligomers at different molecular weights, whose standardization and reproducibility are not so simple.

Only HEK T-Rex cell lysates containing tau with a molecular weight of ~240 kDa, resembling the tetrameric protein, have a toxic effect when administered to *C. elegans*, inducing a motility defect, whose specificity was validated by tau immunodepletion studies. Previous in vitro studies indicated trimeric tau assemblies as the minimal unit responsible for toxicity [14,15]. In our conditions, lysates from HEK T-Rex cells induced for 3 days, mainly containing tau trimers, did not have any toxic effect in worms, it cannot be excluded that in vivo higher concentration of trimers or larger tau conformers are required for a proteotoxic effect. The neuromuscular defects with lysates of cells induced for five days were comparable to those obtained when worms were treated with brain homogenates from mice modeling genetic or sporadic forms of tauopathy [21]. These data suggest that multimeric insoluble assemblies similar to those formed in HEK T-Rex 5 days after induction are also present in the brain of P301L mice and TBI mice.

The mechanisms of the neuromuscular dysfunction caused by tau assemblies remain to be elucidated. We hypothesized that similarly to oligomers of other misfolded proteins, tau oligomers too, once ingested by *C. elegans*, can be absorbed by the gut and diffuse in various tissues, affecting neuromuscular function [29,30].

We then investigated whether this cell-worm-based approach can be applied to studies aimed at discovering drugs to interfere with tau oligomeric toxicity. *C. elegans* has already been proposed as a valuable in vivo model for screening compounds against different protein misfolding diseases [19,30] and the observations from these studies have already been translated into clinical applications [19,31].

As a prototype anti-amyloidogenic compound, we used Doxy, which inhibits the aggregation and oligomerization of different misfolded proteins in vitro and in vivo [32,33,34,35,36,37]. Recent findings with in vitro recombinant tau indicate that this compound can interfere with the aggregation of the protein, the exposure of hydrophobic residues, and cell toxicity [26].

Our findings indicate for the first time that Doxy can also affect tau solubility and promotes the disassembly of already formed toxic aggregates ex vivo. These effects, which translated into a dose-dependent protective action in *C. elegans*, suggest that tetracyclines could be potential compounds for the treatment of tauopathies. In a strategy of “drug repurposing”, this old class of compounds offers promising safe, and inexpensive therapy, backed by data demonstrating their ability to pass the blood–brain barrier [38]. The ability of tetracyclines to interact with oligomers of different amyloidogenic proteins, including amyloid β and tau, is an added value and not a limit, particularly if the idea of the conformation-dependent rather than sequence-dependent role of the protein is considered relevant for the onset of central amyloidogenic disease.

## 4. Materials and Methods

### 4.1. HEK T-Rex Cells

Human embryonic kidney (HEK) 293 cells were engineered to obtain HEK T-Rex cells expressing human tau P301L (Tau P301L) tagged with human influenza hemagglutinin (HA), under the control of a Tet-promoter [9]. Cells were cultivated in Dulbecco’s Modified Eagle Medium (DMEM) with high glucose and pyruvate) (Gibco, Merck, Italy) containing 10% tetracycline-free Fetal Bovine Serum (FBS), 1% Glutamax (Gibco, Merck, Italy), 5 µg/mL blasticidin (Gibco, Merck, Italy), and 100 µg/mL hygromycin B (Invitrogen, Monza, Italy company).

To induce the stable expression of Tau P301L, HEK T-Rex cells were plated in 12-well plates (Corning^®®^ Costar^®®^ TC-Treated Multiple Well Plates, Merck, Italy) at 0.1 × 10^6^ cells/well and after 24 h were treated with 1.0 µg/mL doxycycline Hyclate (Sigma Aldrich, Italy) diluted in DMEM (Induced cells). Control cells were treated with the same volume of DMEM (Not-induced cells). Cells were then incubated at 37 °C in 5% CO_2_ in the air, and 3 and 5 days later cells were harvested in 10 mM phosphate-buffered saline (PBS), centrifuged at 150× *g* for 5 min at 4 °C; the pellet was collected and stored at −80 °C until use.

Experiments were also run to assess the effect of tau expression on cell viability. HEK T-Rex cells, 24 h after plating in 12-well plates at 0.1 × 106 cells/well, were treated with 1.0 µg/mL Doxy diluted in DMEM (Induced cells), and cell viability was determined 1, 3, and 5 days later in MTT and lactate dehydrogenase (LDH) assays. The MTT assay was carried out with thiazolyl blue tetrazolium bromide (Sigma Aldrich, Milan, Italy). The release of LDH was determined on 25 µL of cell media (Sigma Aldrich, Italy).

Cell viability was assessed by incubating cells with 1:10 Alamar Blue reagent (Invitrogen, Italy) in a culture medium (without FBS) for 2 h at 37 °C. Fluorescence was then measured with a spectrofluorimeter (TECAN plate reader, Infinite M200, TECAN, Männedorf, Switzerland) using λ excitation 560 nm and λ emission 590 nm. Not-induced cells treated with DMEM and collected at the same time points were used as controls.

### 4.2. Immunochemical Analysis

Cell pellets were homogenized in 50 µL of 50 mM Tris-HCl, pH 7.5, containing 50 mM KCl and 10 mM MgCl_2_). Protein concentration was quantified with the Pierce BCA Protein Assay kit (Life Technologies, Monza, Italy). Samples were analyzed by immunoblotting using 10% SDS-PAGE gel and Western blotting. After heating the samples at 95 °C for 10 min in a sample buffer containing 5% β-mercaptoethanol (1:1 *v*/*v*, Bio-Rad, Milan, Italy), 30 μg of the proteins were loaded in each lane of the gel. The membranes were blocked with 10 mM Tris-HCl solution, pH 7.5, containing 100 mM NaCl, 0.1% (*v*/*v*) Tween 20, 5% (*w*/*v*) low-fat dry milk powder, and 2% (*w*/*v*) bovine serum albumin, and incubated overnight with the anti-human Tau rabbit polyclonal antibody (1:1000, DAKO, Glostrup, Denmark), anti-HA tag rabbit polyclonal antibody (1:2000, Abcam, Cambridge, UK), anti-oligomeric tau rabbit monoclonal antibody T22 (1:20,000, Merck, Milan, Italy), anti-phospho tau antibody anti-tau paired 198, 199, 202, 205 (1:2000, Abcam) or anti-vinculin mouse monoclonal antibody (1:5000, Merck). Anti-mouse IgG peroxidase conjugate (1:10,000, Sigma Aldrich, Milan, Italy) and anti-rabbit IgG peroxidase conjugate (1:10,000, Sigma Aldrich) were used as secondary antibodies. Recombinant human tau-441 isoform (h-tau 41, Abcam) was used as the standard.

To characterize protein aggregates samples were diluted in a loading buffer composed of 25 mM Tris solution containing 200 mM glycine, 0.2% SDS, 5% glycerol, and 0.025% bromophenol blue, and 30 μg of proteins were loaded in each lane of 2.5% stacking and 6% resolving polyacrylamide gels (semi-denaturing gel). In addition, samples were diluted in 62.5 M Tris solution containing 40% glycerol and 0.001% bromophenol blue, and 30 μg of proteins were loaded into NuPAGE™ 3 to 8% Tris-Acetate, 1.0–1.5 mm, Mini Protein Gels (Thermo Fisher Scientific) (Native gel). At the end of electrophoresis gels were blotted onto the Polyvinylidene fluoride (PVDF) membrane, blocked as described before, and incubated overnight with the anti-human Tau rabbit polyclonal antibody (1:1000), anti-oligomeric tau rabbit monoclonal antibody T22 (1: 20,000) or anti-vinculin mouse monoclonal antibody (1:5000). Anti-mouse IgG peroxidase conjugate (1:10,000) and anti-rabbit IgG peroxidase conjugate (1:10,000) were used as secondary antibodies.

The mean volumes of immunoreactive bands were recorded using Image Lab™ software (Bio-Rad). The data were expressed as the mean of the immunoreactive bands/volume of total vinculin-stained proteins in the spot ± SD.

### 4.3. Detergent Insolubility Assay

Cell lysates were analyzed with an adapted detergent insolubility assay [39]. Briefly, cell lysates (50 µg) were incubated for 20 min at 4 °C in 50 mM Tris-HCl solution, pH 7.5, containing 0.5% Triton X-100, 0.5% NP-40 and 0.5% sodium deoxycholate. Samples were centrifuged at 150× *g* at 4 °C for 5 min, the supernatants were collected and centrifuged at 100,000× *g* at 4 °C for 50 min. The supernatant (Soluble fraction) and the pellet (Insoluble fraction) were collected and analyzed in 10% SDS-Page gel. The soluble fraction was heated at 95 °C for 10 min in 0.5 M Tris-HCl solution, pH 6.8, containing 10% SDS, 12% β-2-mercaptoethanol, and 50% glycerol and 0.001% bromophenol blue, and 50 μg of proteins were loaded in each gel lane. The insoluble fraction was suspended in 1 mL of 10 mM PBS and centrifuged at 50,000× *g* at 4 °C for 30 min. The supernatant was collected and heated at 95 °C for 10 min in 1 M Tris-HCl solution, pH 6.8, containing 20% SDS, 24% β-2-mercaptoethanol, 50% glycerol, and 0.001% bromophenol blue, and 50 μg of proteins were loaded in each gel lane. At the end of electrophoresis gels were blotted onto PVDF membrane, blocked with 10 mM Tris-HCl solution, pH 7.5, containing 100 mM NaCl, 0.1% Tween 20, 5% low fat dry milk powder, and 2% bovine serum albumin, and incubated overnight with anti-human Tau rabbit polyclonal antibody (1:1000) or anti-vinculin mouse monoclonal antibody (1:5000). Peroxidase-conjugated anti-mouse and anti-rabbit IgG (1:10,000) were used as secondary antibodies. The mean volumes of immunoreactive bands were quantified as described before. The data were expressed as the mean of the immunoreactive bands/volume of total vinculin-stained proteins in the spot ± SD.

### 4.4. Tau Immunoprecipitation

Cell lysates were diluted in 10 mM PBS, pH 7.4, containing 0.1% TritonX-100, to obtain a solution of 10 µg of proteins/100 µL. Samples were then incubated overnight at 4 °C under orbital shaking, with 1 µg of anti-HA tag mouse monoclonal antibody (Enzo Life Sciences Inc., Lausen, Switzerlandy**)** or anti-Green Fluorescent Protein (GFP) mouse monoclonal antibody (B-2, Santa Cruz Biotechnology Inc., Heidelberg Germany). At the end of incubation, samples were mixed with 20 µL of Protein A and protein G Resin beads (Genespin Srl, Milan, Italy), then incubated for 2 h at 4 °C under orbital shaking. Samples were centrifuged at 150× *g* for 5 min at 4 °C, the supernatant was collected and centrifuged again at 100× *g* for 5 min at 4 °C. The supernatants were then analyzed by SDS-PAGE and Western blot to determine the amount of tau in immunoprecipitated samples (output). Thirty µg of cell lysates were analyzed in the same experimental condition to determine the level of tau before immunoprecipitation (input). Twenty µL of input and output samples were analyzed as described before and incubated overnight with anti-human Tau rabbit polyclonal antibody (1:1000) or anti-vinculin mouse monoclonal antibody (1:5000). Peroxidase-conjugated anti-mouse and anti-rabbit IgG (1:10,000) were used as secondary antibodies. Input and output samples were also employed for *C. elegans* experiments.

### 4.5. C. elegans Studies

Bristol N2 nematodes were obtained from the *Caenorhabditis elegans* Genetic Center (CGC, University of Minnesota, Minneapolis, MN, USA) and propagated at 20 °C on solid Nematode Growth Medium (NGM) seeded with *E. coli* OP50 (CGC) for food. Age-synchronized animals were obtained by the bleaching technique [40]. *C. elegans* at the first larval stage were transferred to fresh NGM plates and grown at 20 °C. At L3-L4 larval stage nematodes were collected with 10 mM PBS, pH 7.4, centrifuged, and washed twice with PBS to eliminate bacteria. Worms were incubated for 2 h at room temperature with orbital shaking, in the absence of *E. coli*, with lysates of Not-induced or Induced cells (30 µg protein/100 worms/100 µL of 10 mM PBS, pH 7.4). Control worms were incubated with 10 mM PBS, pH 7.4 (100 worms/100 µL).

Worms were then plated onto NGM plates seeded with OP50 *E. coli*, grown at 20 °C, and transferred every day for six days to new NGM plates seeded with *E. coli* to avoid overlapping generations [21]. The locomotor activity of nematodes was scored seven days after the treatment by counting the number of left-right movements in one minute in liquid (body bends/min) [41,42]. All behavioral evaluations were performed blinded.

In similar experiments, worms were incubated with cell lysates previously immunoprecipitated with anti-HA tag mouse monoclonal antibody or anti-GFP mouse monoclonal antibody as previously described (30 µg protein/100 worms/100 µL). As negative controls, anti-HA tag mouse monoclonal antibody and anti-GFP mouse monoclonal antibodies were administered alone.

### 4.6. Effect of Doxy

To investigate the effect of Doxy on the toxicity, worms at the L3-L4 larval stage were fed for 2 h at room temperature with orbital shaking, in the absence of *E. coli*, with cell lysates of HEK T-Rex cells induced for 5 days (30 µg protein/100 worms/100 µL of 10 mM PBS, pH 7.4) containing or not 0–200 μM doxycycline Hyclate (Sigma Aldrich, Milan, Italy). Control worms were incubated with 10 mM PBS, pH 7.4 (100 worms/100 µL). At the end of incubation, worms were plated onto NGM plates seeded with OP50 *E. coli*, grown at 20 °C, and transferred every day for 6 days to new NGM plates seeded with *E. coli* to avoid overlapping generations. The locomotor activity of nematodes was scored 7 days after the treatment as described before.

Lysates of HEK T-Rex cells (30 µg protein/100 µL of 10 mM PBS, pH 7.4) incubated or not with 50 µM Doxy for 2 h at 20 °C in orbital shacking, were analyzed through detergent insolubility assay or semi-denaturing gels as described before. Membranes were blotted with anti-human Tau rabbit polyclonal antibody (1:1000) or anti-vinculin mouse monoclonal antibody (1:5000). Peroxidase-conjugated anti-mouse and anti-rabbit IgG (1:10,000) were used as secondary antibodies

### 4.7. Statistical Analysis

The data were analyzed using GraphPad Prism 8.0 software (San Diego, CA, USA) by Student’s t-test, one-way or two-way ANOVA, and Bonferroni’s or Tukey’s post hoc test. A *p*-value < 0.05 was considered significant.

## Figures and Tables

**Figure 1 ijms-23-11277-f001:**
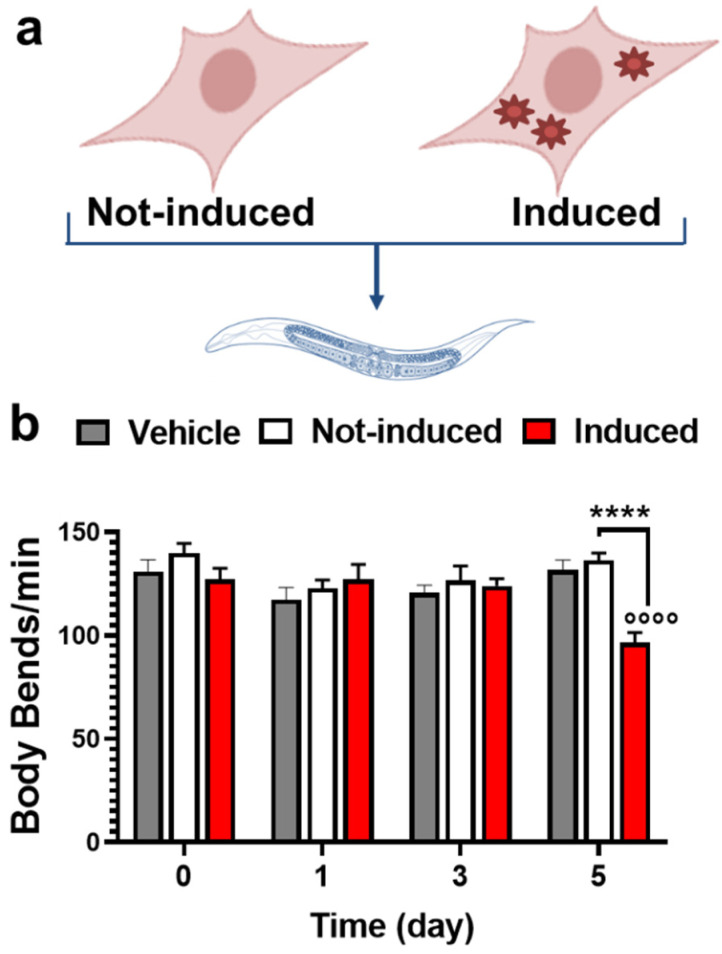
**Time-dependent effect of the induction of tau P301L expression in HEK T-Rex cells on *C. elegans* motility.** (**a**) HEK T-REx cells were treated with 1.0 µg/mL Doxy to induce the expression of human tau P301L (Induced) or the same volume of 10 mM PBS, pH 7.4 (Not-induced). (**b**) Cell lysates were prepared from cells collected immediately after (0) and 1, 3, or 5 days after the treatment and administered to *C. elegans* at the final concentration of 30 μg proteins/100 worms/100 μL. As controls, worms were treated in the same experimental conditions with 10 mM PBS, pH 7.4 (Vehicle) (100 worms/100 μL). Locomotor activity was recorded 7 days after treatment. Data are mean ± SEM (N = 50 worms/group). °°°° *p* < 0.0001 vs. Vehicle and **** *p* < 0.0001 vs. Not-induced, one-way ANOVA and Bonferroni’s post hoc test.

**Figure 2 ijms-23-11277-f002:**
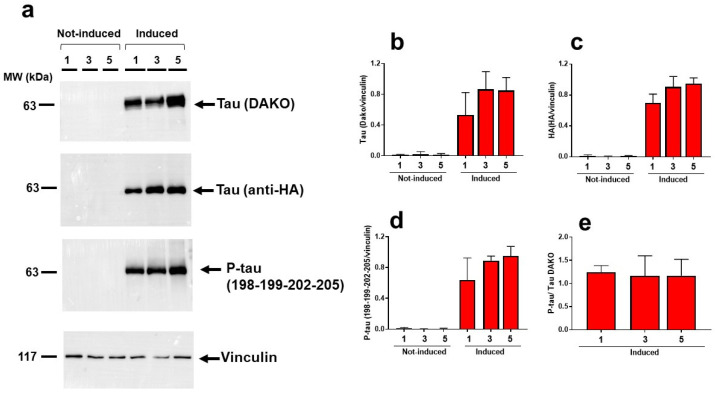
**Time-dependent effect of the induction of HEK T-REx cells on the expression and phosphorylation of tau.** (**a**) Representative Western blot of total tau and phosphorylated tau (P-Tau) in lysates of HEK T-Rex cells collected 1, 3, or 5 days after treatment with doxycycline (Induced) or 10 mM PBS, pH 7.4 (Not-induced). Equal amounts of proteins were loaded in each gel lane (30 μg) and immunoblotted with anti-tau (DAKO), anti-HA tag, anti-P-tau (198-199-202-205), or anti-vinculin antibody. (**b**,**c**) Total tau quantification is expressed as the mean volume of (**b**) DAKO and (**c**) HA tag band immunoreactivity/vinculin. (**d**) P-tau band immunoreactivity/vinculin band immunoreactivity. Data are mean ± SD (N = 3). (**e**) The ratio of the immunoreactivity signal of P-tau/vinculin to Tau DAKO/vinculin of Induced cells. Data are the mean volume of the immunoreactive band/Vinculin ± SD from three independent experiments.

**Figure 3 ijms-23-11277-f003:**
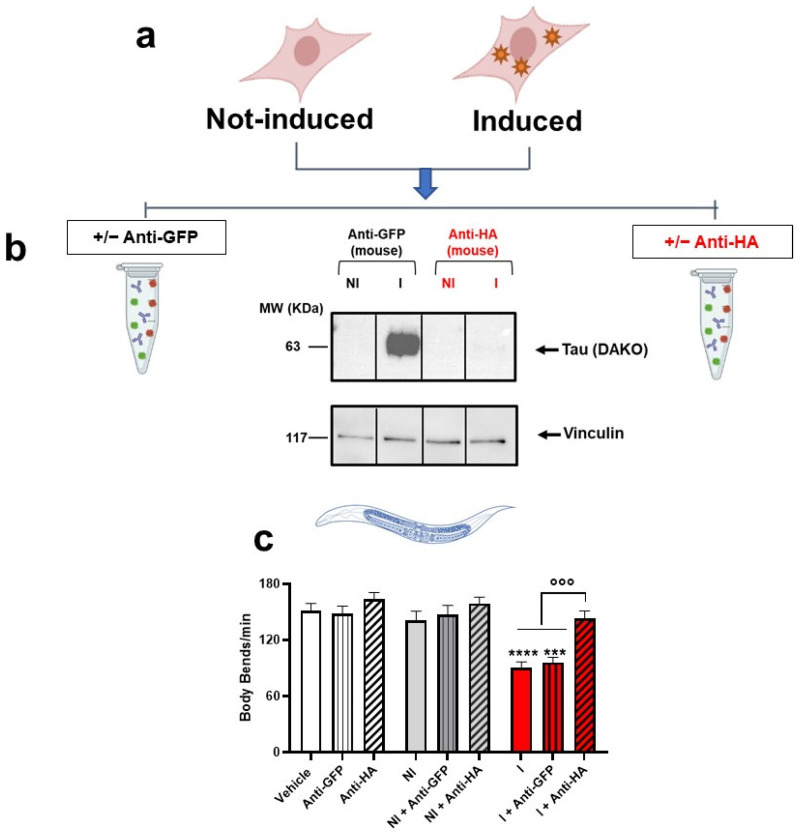
**Tau immunoprecipitation abolishes toxicity.** (**a**) Lysates from HEK T-REx cells (10 μg proteins) induced (I) or not induced (NI) for 5 days were immunoprecipitated with (**b**) 1 μg of an anti-GFP mouse monoclonal antibody (+GFP) or anti-HA tag mouse monoclonal antibody (+ anti-HA). Tau in post-immunoprecipitation lysates was measured by Western blot analysis. Equal amounts of cell lysates (20 μL) were loaded in each gel lane and immunoblotted with an anti-tau antibody (DAKO) or anti-vinculin antibody. (**c**) The immunoprecipitated (+anti-GFP or + anti-HA) or untreated cell lysates were administered to worms (50 μL cell lysate/50 worms). Control worms were treated in the same experimental conditions (50 μL/50 worms) with 10 mM PBS, pH 7.4 (Vehicle), anti-HA tag mouse monoclonal antibody, or anti-GFP mouse monoclonal antibody. Locomotor activity was recorded 7 days after treatment. Data are mean ± SEM (N = 10 worms/group). *** *p* = 0.0003 and **** *p*< 0.0001 vs. Vehicle, °°° *p* = 0.0001 vs. Induced and Induced + anti-GFP, one-way ANOVA and Bonferroni’s post hoc test.

**Figure 4 ijms-23-11277-f004:**
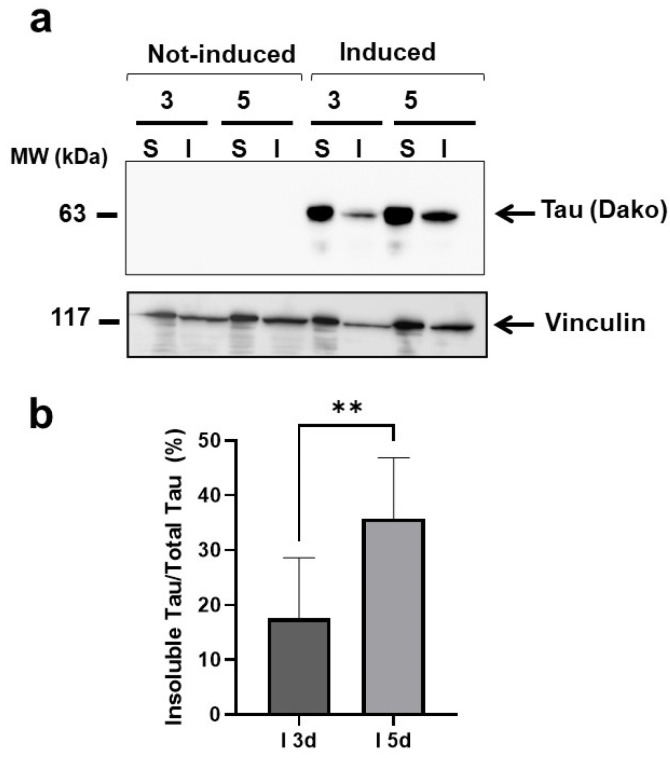
**Detergent insolubility assay of tau P301L produced by HEK T-Rex cells 3 and 5 days after induction.** Lysates were prepared from HEK T-Rex cells collected 3 or 5 days after induction (Induced) or treated with 10 mM PBS, pH 7.4 (Not-induced). (**a**) Representative Western blotting showing the detergent insolubility assay of soluble (S) and insoluble (I) fractions probed with anti-tau DAKO antibody or anti-vinculin antibody. (**b**) Tau quantification in the S and I fractions are expressed as the mean percentage immunoreactivity of the DAKO signal in the insoluble fraction/total Tau (soluble + insoluble fraction). Data are mean ± SD (N = 8) ** *p* < 0.01, one-way ANOVA, and Bonferroni’s post hoc test.

**Figure 5 ijms-23-11277-f005:**
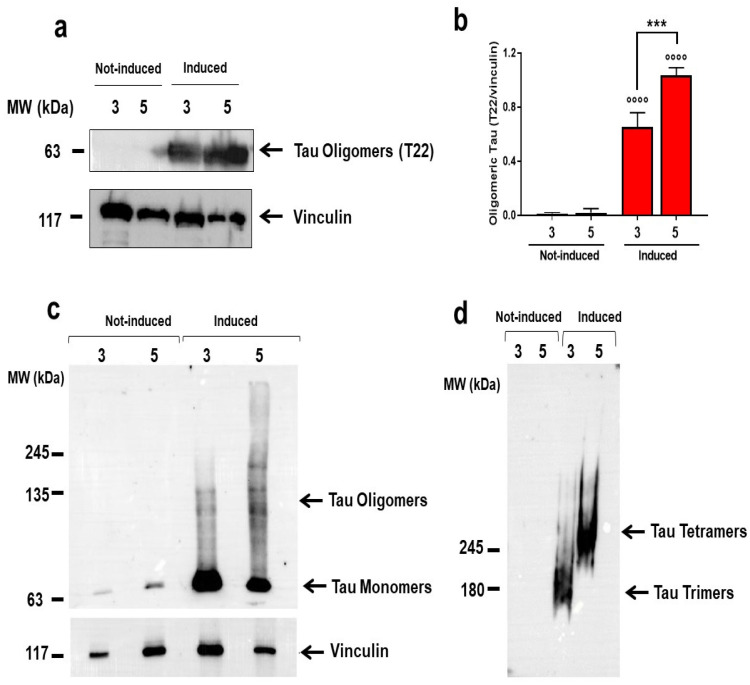
**Tau P301L oligomers in HEK T-Rex induced cells.** (**a**) Representative Western blot of oligomeric tau in lysates of HEK T-Rex cells collected 3 or 5 days after treatment with doxycycline (Induced) or 10 mM PBS, pH 7.4 (Not-induced). An equal amount of proteins were loaded in each gel lane (30 μg) and immunoblotted with anti-oligomeric tau (T22) or anti-vinculin antibody. (**b**) Oligomeric tau quantification expressed as the mean volume of the T22 band immunoreactivity/vinculin. Data are mean ± SD (N = 3). °°°° *p* < 0.0001 vs. Not-induced at the corresponding time point and *** *p* < 0.0005, one-way ANOVA and Bonferroni’s post hoc test. (**c**,**d**) Equal amounts of proteins (30 µg) of cell lysates were loaded in each lane of (**c**) semi-denaturing gel or (**d**) native PAGE 3–8% gradient gel and immunoblotted with anti-tau DAKO antibody or anti-vinculin antibody.

**Figure 6 ijms-23-11277-f006:**
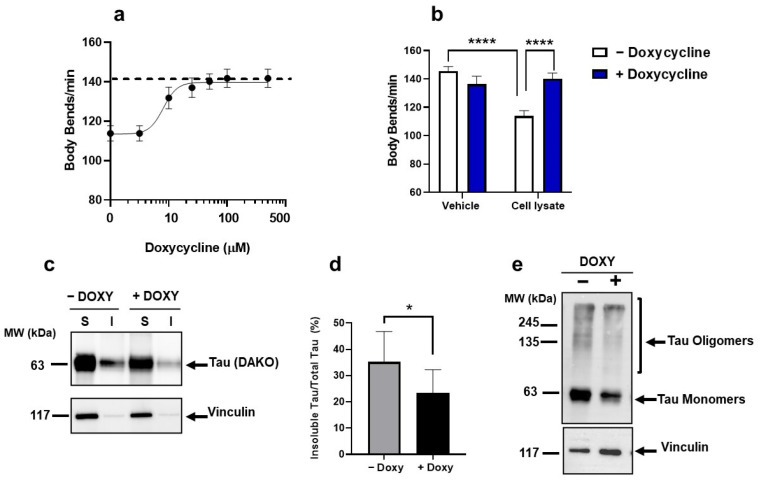
**Doxy protected from the toxicity induced by oligomeric tau.** (**a**) Dose–response effect of Doxy on the locomotor dysfunction induced in worms by lysates of HEK T-Rex cells induced for five days. Worms were fed for 2 h with 30 μg of cell lysates previously incubated for 2 h with 0–200 μM doxycycline. Control worms received the same volume of 10 mM PBS, pH 7.4 alone (dashed line). Locomotor activity was determined 7 days after the treatment. Each value is the mean  ±  SEM, N  =  40. (**b**) Body bends of worms treated with lysates of HEK T-Rex cell induced for 5 days (Cell lysate) or 10 mM PBS, pH 7.4 (Vehicle) in the presence (blue bars) or absence (white bars) of 50 μM Doxy. Locomotor activity was determined 7 days after the treatment. Data are the mean  ±  SEM, N =  45. **** *p* < 0.0001, one-way ANOVA and Bonferroni’s post hoc test. Interaction cell lysate/Doxy= *p* < 0.0001, two-way ANOVA and Bonferroni’s post hoc test. (**c**) Representative Western blotting of the detergent insolubility assay of soluble (S) and insoluble (I) fractions of lysates of cells induced for 5 days and incubated for 2 h in the absence (−DOXY) or presence of 50 μM doxycycline (+DOXY) and probed with anti-tau DAKO antibody or anti-vinculin antibody. (**d**) Tau quantification in the S and I fractions is expressed as the mean immunoreactivity of the DAKO signal in the insoluble fraction/total Tau (soluble + insoluble fractions) immunoreactivity. Data are mean ± SD (N = 8 * *p* < 0.05, Student’s T test. (**e**) Equal amounts of proteins (30 µg) of lysate of cells induced for 5 days and incubated for 2 h without (−DOXY) or with 50 μM doxycycline (+DOXY) were loaded in each lane of semi-denaturing gel and immunoblotted with anti-tau DAKO antibody or anti-vinculin antibody.

## Data Availability

Data supporting reported results can be requested from the corresponding authors.

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
