# Peer review of "A Combined Cell-Worm Approach to Search for Compounds Counteracting the Toxicity of Tau Oligomers In Vivo"

_ijms, 2022, doi:10.3390/ijms231911277_

Round 1

Reviewer 1 Report

In this manuscript, tau protein induced from HEK293 cells was used to treat C. elegans to evaluate its effect on worms’ locomotor activity. By correlating the aggregation/phosphorylation state of tau protein to compromised mobility of C. elegans, this study developed a cell-worm approach to screen compounds for resolving tau toxicity. It is an interesting story and a novel platform for potential pre-pharmatheutical application. However, I have found some caveats that in experiment design and data exhibition. They are listed as follows:

1.    Why were kidney cells (HEK293) selected for tau expression, not any types of neural cell lines?

2.    How did you quality-control induced/non-induced cell lysate? Were there other differences besides w/o tau aggregates?

3.    Fig 1b: There were three groups of C. elegans when testing locomotor: vehicle, non-induced cell lysate, and induced cell lysate. However, the figure was not presenting the numbers from the vehicle group. Instead, a dashed line was presented for basal movement. It is more reasonable to use actually numbers from vehicle group, as the numbers of body bends could not be the exactly same across vehicle groups.

4.    Fig 2: The worms were treated with cell lysate from time 0, 1, 3, and 5 days as shown in Fig 1, but the western blots of time 0 was missing in Fig 2.

5.    Fig 3: Why was anti-GFP antibody used for negative control in anti-HA immunoprecipitation? The anti-HA was from rabbit while anti-GFP was from mouse. They were not good pair for experiment/control.  The background of Vinculin blot is too dark to view the actual bands.

6.    Fig 3c: vehicle plus anti-HA/non-specific antibody should be added to serve as a complete set of control to eliminate the effect of antibodies.

Author Response

Response to Reviewer 1 Comments

Q1.    Why were kidney cells (HEK293) selected for tau expression, not any types of neural cell lines?

A1. HEK-293 cells were selected because, due to their efficient transfection of plasmid DNAs, faithful translation, and processing of proteins, are widely used as a heterologous expression system for expressing recombinant proteins. In addition, HEK T-Rex cells expressing tau P301L tagged with a green fluorescent protein (GFP) have been used as a cell-based tau overexpression assay to screen for pharmacological compounds against tau aggregation (ref).

Q2.    How did you quality-control induced/non-induced cell lysate? Were there other differences besides w/o tau aggregates?

A2. Our paper aimed to correlate the level of expression of tau and its conformational state with proteotoxicity. For this reason, we focused on tau and P-tau levels and did not do a full characterization of the cell lysate composition of induced and non-induced cells. We also evaluated the effect of the induction on the cell viability and, as reported in Figure S1, no differences between the various experimental groups were observed. In the cell lysates prepared by non-induced cells and the cells induced for different times we also determined the levels of actin and vinculin, two proteins important for the integrity of the cell cytoskeleton. As reported in Figure 2 and Figure S2, the induction of tau expression did not cause any change in the levels of these two proteins. These data, combined with the fact that the immunoprecipitation of tau present in lysates of cells induced for 5 days reverts their ability to induce motor dysfunction in C. elegans, indicates that tau plays a key role in toxicity.

Q3.    Fig 1b: There were three groups of C. elegans when testing locomotor: vehicle, non-induced cell lysate, and induced cell lysate. However, the figure was not presenting the numbers from the vehicle group. Instead, a dashed line was presented for basal movement. It is more reasonable to use actually numbers from vehicle group, as the numbers of body bends could not be the exactly same across vehicle groups.

A3. As requested, Figure 1b has been modified and the data of vehicle groups have been added.

Q4.    Fig 2: The worms were treated with cell lysate from time 0, 1, 3, and 5 days as shown in Fig 1, but the western blots of time 0 was missing in Fig 2.

A4. A Western blot showing the level of tau and P-tau in cell lysates at time 0 has been added in the Supplementary Materials as Figure S2. In the Results section, on page 4, line 132, the sentence describing the data in Figure 2 has been modified as follows: “As shown in Figure 2, similar amounts of tau and P-tau were produced in Induced cells from 1 to 5 days of induction whereas no tau and P-tau were detected in at time 0 (Fig. S2), indicating that neither the level of the protein nor its degree of phosphorylation correlated with the toxicity in the worms”.

Q5.    Fig 3: Why was anti-GFP antibody used for negative control in anti-HA immunoprecipitation? The anti-HA was from rabbit while anti-GFP was from mouse. They were not good pair for experiment/control. The background of Vinculin blot is too dark to view the actual bands.

A5. An anti-GFP antibody was used as a negative control because it does not recognize tau and HA and is thus unable to bind to both of them. The fact that the anti-HA and anti-GFP antibodies were generated one in the rabbit and the other in the mouse is not relevant for their specificity. For this reason, we are convinced that the use of anti-GFP as a negative control was correct. As requested, to make the vinculin bands more visible we have lightened the Western blot trying to reduce the background.

Q6.    Fig 3c: vehicle plus anti-HA/non-specific antibody should be added to serve as a complete set of control to eliminate the effect of antibodies.

A6. We are sorry but unfortunately, this further check has not been done. We hope that the reviewer does not judge the lack of this data as negative, also because the results obtained with the immunoprecipitation of non-induced cells are itself a control.

Reviewer 2 Report

Here the authors establish a novel system to access the toxicity of oligomeric tau assemblies. Aggregation prone P301L tau protein was expressed in HEK293 cells under the control of a tetracycline-inducible promoter and fed to C. elegans worms. The authors convincingly show that this tau protein impaired the worm's motility. They further examined the oligomeric state of the tau protein and presented evidence that the toxic species corresponds to a tau tetramer. Preincubation of the tau lysate with doxycycline reduced its toxicity by altering the oligomeric state of tau. This system therefore seems to be perfectly suited for pharmacological screening of compounds against tau toxicity.

The data presented are convincing and the necessary controls were properly performed. Although, it is still enigmatic how the ingested tau oligomers affect the worm's motility, the results are very interesting and warrant publication in IJMS. 

Author Response

Response to Reviewer 2 Comments

Q1. Here the authors establish a novel system to access the toxicity of oligomeric tau assemblies. Aggregation prone P301L tau protein was expressed in HEK293 cells under the control of a tetracycline-inducible promoter and fed to C. elegans worms. The authors convincingly show that this tau protein impaired the worm's motility. They further examined the oligomeric state of the tau protein and presented evidence that the toxic species corresponds to a tau tetramer. Preincubation of the tau lysate with doxycycline reduced its toxicity by altering the oligomeric state of tau. This system therefore seems to be perfectly suited for pharmacological screening of compounds against tau toxicity.

The data presented are convincing and the necessary controls were properly performed. Although, it is still enigmatic how the ingested tau oligomers affect the worm's motility, the results are very interesting and warrant publication in IJMS. 

A1. We thank the reviewer for the positive comments on our work.

Round 2

Reviewer 1 Report

The authors did not fully address my concerns on experimental design and data exhibition. The proper IgG served as negative control in immunoprecipitation is key to confident results. However, the revision is not aware of that. The missing control in Fig 3 is a big caveat as we can not exclude the possibility that antibody alone contributes to the results. Again, the story is interesting but some of the experiments should be re-done to draw more solid conclusions. 

Author Response

Q1. The authors did not fully address my concerns on experimental design and data exhibition. The proper IgG served as negative control in immunoprecipitation is key to confident results. However, the revision is not aware of that. The missing control in Fig 3 is a big caveat as we can not exclude the possibility that antibody alone contributes to the results. Again, the story is interesting but some of the experiments should be re-done to draw more solid conclusions. 

A1. We performed new immunoprecipitation experiments employing an anti-HA tag and anti-GFP antibodies both generated in the mice, thus using as requested, the proper IgG as the negative control. The results obtained, shown in the new Figure S4, were superimposable with those already reported in Figure 3 of the manuscript, and indicated that the tau removal abolished the toxic effect observed in C. elegans. Accordingly, the description of the data, in the Results section, on page 4, was modified as follows: “Lysates were analyzed by Western blot before (Fig. S3 and Fig. S4a) and after the immunoprecipitation to confirm tau depletion (Fig. 3a-b and Fig. S4b) and then given to C. elegans. Tau immunodepletion abolished the motility defect caused by the lysates from cells induced for 5 days (Fig. 3c and Fig. S4c), indicating that tau had a key role in the toxic effect in C. elegans.

The Materials and Methods section, on pages 12 and 13, was modified detailing the new antibody used and the C. elegans experiments performed for the revision.

Round 3

Reviewer 1 Report

The authors have greatly improved on experimental design and presentation on Figure 3. The new data in Fig S4 is a replicate of Fig 3 but with high quality and soundness. I would suggest to replace Fig 3 by Fig S4. 

Author Response

Q1. The authors have greatly improved on experimental design and presentation on Figure 3. The new data in Fig S4 is a replicate of Fig 3 but with high quality and soundness. I would suggest to replace Fig 3 by Fig S4.

A1. As requested by the Reviewer, Figure 3 has been replaced by Figure S4, and the text modified accordingly.